# Two Antagonistic Microtubule Targeting Drugs Act Synergistically to Kill Cancer Cells

**DOI:** 10.3390/cancers12082196

**Published:** 2020-08-06

**Authors:** Lauralie Peronne, Eric Denarier, Ankit Rai, Renaud Prudent, Audrey Vernet, Peggy Suzanne, Sacnicté Ramirez-Rios, Sophie Michallet, Mélanie Guidetti, Julien Vollaire, Daniel Lucena-Agell, Anne-Sophie Ribba, Véronique Josserand, Jean-Luc Coll, Patrick Dallemagne, J. Fernando Díaz, María Ángela Oliva, Karin Sadoul, Anna Akhmanova, Annie Andrieux, Laurence Lafanechère

**Affiliations:** 1INSERM U1209, CNRS UMR5309, Team Regulation and Pharmacology of the Cytoskeleton, Department of Microenvironment, Cell Plasticity and Signaling, Institute for Advanced Biosciences, Université Grenoble Alpes, 38000 Grenoble, France; lauralie.peronne@umontreal.ca (L.P.); renaud_prudent@yahoo.fr (R.P.); vernet_a@yahoo.fr (A.V.); sacnicte.ramirez-rios@univ-grenoble-alpes.fr (S.R.-R.); sophie.michallet@univ-grenoble-alpes.fr (S.M.); anne-sophie.ribba@univ-grenoble-alpes.fr (A.-S.R.); karin.sadoul@univ-grenoble-alpes.fr (K.S.); 2Grenoble Institute of Neurosciences, INSERM U1216, Université Grenoble Alpes, CEA, 38000 Grenoble, France; eric.denarier@univ-grenoble-alpes.fr (E.D.); annie.andrieux@univ-grenoble-alpes.fr (A.A.); 3Cell Biology, Neurobiology and Biophysics, Department of Biology, Faculty of Science, Utrecht University, 3584 CH Utrecht, The Netherlands; A.Rai@uu.nl (A.R.); a.akhmanova@uu.nl (A.A.); 4Normandie Univ., UNICAEN, CERMN, 14032 Caen, France; peggy.suzanne@unicaen.fr (P.S.); patrick.dallemagne@unicaen.fr (P.D.); 5INSERM U1209, CNRS UMR5309, Team Cancer Targets and Experimental Therapeutics, Department of Microenvironment, Cell Plasticity and Signaling, Institute for Advanced Biosciences, Université Grenoble Alpes, 38000 Grenoble, France; melanie.guidetti@univ-grenoble-alpes.fr (M.G.); julien.vollaire@univ-grenoble-alpes.fr (J.V.); veronique.josserand@ujf-grenoble.fr (V.J.); jean-luc.coll@univ-grenoble-alpes.fr (J.-L.C.); 6Structural and Chemical Biology Department, Centro de Investigaciones Biológicas, CSIC, Ramiro de Maeztu 9, 28040 Madrid, Spain; lucena@cib.csic.es (D.L.-A.); fer@cib.csic.es (J.F.D.); marian@cib.csic.es (M.Á.O.)

**Keywords:** cancer therapy, microtubules, drug synergy, carbazole, paclitaxel

## Abstract

Paclitaxel is a microtubule stabilizing agent and a successful drug for cancer chemotherapy inducing, however, adverse effects. To reduce the effective dose of paclitaxel, we searched for pharmaceutics which could potentiate its therapeutic effect. We screened a chemical library and selected Carba1, a carbazole, which exerts synergistic cytotoxic effects on tumor cells grown in vitro, when co-administrated with a low dose of paclitaxel. Carba1 targets the colchicine binding-site of tubulin and is a microtubule-destabilizing agent. Catastrophe induction by Carba1 promotes paclitaxel binding to microtubule ends, providing a mechanistic explanation of the observed synergy. The synergistic effect of Carba1 with paclitaxel on tumor cell viability was also observed in vivo in xenografted mice. Thus, a new mechanism favoring paclitaxel binding to dynamic microtubules can be transposed to in vivo mouse cancer treatments, paving the way for new therapeutic strategies combining low doses of microtubule targeting agents with opposite mechanisms of action.

## 1. Introduction

Microtubules (MTs), dynamic polymeric filaments composed of α-tubulin and β-tubulin heterodimers, are key components of the cytoskeleton of eukaryotic cells. Their crucial roles in cell division and physiology mainly rely on their ability to rapidly polymerize or depolymerize. Targeted perturbation of this finely tuned process constitutes a major therapeutic strategy. Drugs interfering with MTs are major constituents of chemotherapies for the treatment of carcinomas. A number of compounds bind to the tubulin-MT system. They can be roughly classified into MT-stabilizers such as taxanes or epothilones, and MT-destabilizers such as vinca alkaloids, combretastatin, and colchicine [1]. It has been demonstrated that binding of vinca alkaloids or colchicine prevents the curved-to-straight conformational change of tubulin at the tip of the growing MT, necessary for proper incorporation of new tubulin dimers into the MT lattice (see reviews [1,2]).

Paclitaxel (PTX) binds to the taxane-site of β-tubulin and stabilizes the MT lattice by strengthening lateral and/or longitudinal tubulin contacts within the MT [1]. At stoichiometric concentrations, it promotes MT assembly. At low and clinically relevant concentrations, PTX primarily suppresses MT dynamics without significantly affecting the MT-polymer mass [3,4]. PTX is one of the most successful chemotherapeutic drugs in history. It is currently used to treat patients with a variety of cancers including lung, breast, and ovarian cancers [5].

Several mechanisms have been proposed to explain the anti-tumor activity of PTX. It can induce a mitosis dependent cell death, either by producing a mitotic arrest [6], when applied at high concentrations, or by promoting chromosome mis-segregation at low concentrations [7]. Alternatively, PTX can act on interphase cells and drive autonomous cell death by perturbation of intracellular trafficking [8]. It has also been recently proposed that post-mitotic formation of micronuclei induced by PTX can promote inflammation and subsequent tumor regression via vascular disruption and immune activation [9].

While PTX is a successful anti-cancer drug, its low solubility, its toxicity, and the fact that cells become resistant to this drug, impose serious limits to its use. Cell resistance to PTX is due to the high expression of P-glycoprotein or multidrug resistance-associated proteins, as well as to the overexpression of some β-tubulin isoforms or mutations in β-tubulin that affect the MT polymer mass and/or drug binding [10]. Another major drawback of PTX in clinical applications is the development of peripheral neuropathies, primarily involving the sensory nervous system. Although the molecular bases of these neuropathies are not completely understood, an inhibition of MT-based axonal transport appears to be a possible mechanism [11]. It has been recently shown that anterograde kinesin based-axonal transport is specifically affected by PTX, whereas MT destabilizing drugs that bind preferentially to the ends of MTs have much less effect on axonal transport [12].

An alternative therapeutic solution would be the use of pharmaceutics which, when co-administrated with PTX, could potentiate its effect without significantly increasing its toxicity. Such agents could allow the use of lower doses of PTX in cancer therapy, may limit the occurrence of resistances and reduce MT-independent adverse effects.

To identify such agents, we screened a collection of 8000 original compounds using a cytotoxicity assay and selected a derivative of the carbazole series (Carba1) able to sensitize cells to a low, non-toxic dose of PTX. We demonstrated that Carba1 exerts synergistic cytotoxic effects with PTX. In cells, Carba1 has no major effect on the total MT mass in interphase cells and shows moderate cytotoxicity. We found that Carba1 targets the colchicine-binding site of tubulin, inhibits in vitro tubulin polymerization and promotes catastrophes, similar to other MT-destabilizing agents. A combination of Carba1 and PTX causes synergistic perturbation of MT growth in vitro. Carba1-induced modulation of MT dynamics increases the binding of fluorescent taxane, Fchitax-3 to MTs, similar to what we have described previously for vinblastine [13], providing a biochemical explanation of the observed synergy between Carba1 and PTX.

Carba1 has no major anti-tumor effect when administrated alone in animals and no detectable toxicity. The administration of a combination of Carba1 and a low, ineffective, dose of PTX showed, however, a significant effect on tumor growth, indicating that Carba1 and PTX act synergistically in vivo. Our results pave the way for new therapeutic strategies, based on the combination of low doses of MT targeting agents with opposite mechanisms of action. These combinations may have reduced toxicity compared to high therapeutic PTX doses.

## 2. Results

### 2.1. A Pairwise Chemical Genetic Screen Identifies a Carbazole Derivative, Carba1, That Sensitizes Cells to Paclitaxel

We designed a screen to select compounds that sensitize cells to paclitaxel (PTX). In a first step, we determined a minimal dose of PTX that is not toxic for cells. We found that 1 nM of PTX showed no toxicity when applied on HeLa cells for 48 h (Appendix A). Furthermore, such a dose has no detectable impact on MT dynamics as assessed by EB3 tracking after time lapse fluorescence microscopy using GFP-EB3-transfected HeLa cells and subsequent calculation of dynamic instability parameters (Appendix A).

We then screened a library of 8000 compounds at a concentration of 5 µM (Figure 1A and Appendix A) and compared their cytotoxicity on HeLa cells when administrated alone or in combination with 1 nM PTX. We selected 76 compounds that show no or moderate cytotoxicity when applied alone, and that were found cytotoxic when applied in combination with 1 nM PTX. We observed that many carbazole derivatives were present in this selection (Appendix A) and we decided to focus our study on the 6-chloro-1,4-dimethyl-3-pyrrol-1-yl-9H-carbazole (Carba1, Figure 1B) because it lacked reactive chemical groups that could interact non-specifically with protein targets, was minimally cytotoxic when assayed alone on HeLa cells (Appendix A and Figure 1C and Figure 2A) and showed a synergistic activity with PTX (Figure 1C) when assayed using the sensitive PrestoBlue assay. We further used the combination index (CI) method of Chou-Talalay [14] to quantify the synergistic interaction between Carba1 and PTX. A CI value of less than 1 indicates synergism, a CI value = 1 indicates an additive effect, and CI > 1 indicates antagonism. Interestingly, we found that the synergistic effect was closely dependent on PTX concentrations. Indeed, for all Carba1 concentrations tested, synergism was observed for PTX concentrations lower than 5 nM, additive effects for a PTX concentration of 5 nM, and antagonistic effects for all higher concentrations (7.5 and 10 nM PTX, Appendix A).

A clear synergism (CI = 0.57) was observed for PTX 1 nM plus Carba1 12 µM. At these concentrations, the compounds showed low cytotoxic activity when applied alone. Indeed, the comparison of HeLa cell apoptosis induced by Carba1 (12 μM), PTX (1 nM) to the apoptosis induced by the combination of Carba1 and PTX (12 μM/1 nM) confirmed the synergistic activity (Figure 1D).

### 2.2. Carba1 Has a Moderate Cytotoxicity When Applied at High Concentrations

As our final aim was to test the therapeutic efficacy of Carba1 in combination with PTX, it was important to investigate its cellular effects and to check that this compound is not or moderately toxic by itself. We first analyzed the cytotoxicity of Carba1 on HeLa cells, using the PrestoBlue assay. As shown in Figure 2A, Carba1 has a moderate cytotoxicity with a calculated GI50 (50% of growth inhibition) of 21.8 μM after a 72-h treatment.

Since the Prestoblue assay is a metabolic test that indirectly measures cell viability, we directly detected cells in apoptosis using Annexin V staining, and quantified them by flow cytometry. We compared the effect of two concentrations of Carba1: a concentration (12 μM) that has no detectable effect on cell viability and a cytotoxic concentration (25 μM). No apoptosis was detected when Carba1 was applied for 72 h at a concentration of 12 μM whereas at 25 µM, it induced apoptosis of 30% of the cells (Figure 2B,C).

These results indicate that Carba1 is only weakly toxic, even when applied at a high concentration. A toxicity analysis of a single 10 μM dose of Carba1 on a set of 60 human cancer cell lines (NCI-60 screen [15]) confirmed the low cytotoxic activity of Carba1 (Appendix A).

Finally, we checked that Carba1 was not toxic on a normal cell line. We used immortalized RPE-1 human cells and compared the effect of Carba1, PTX, and their combination on the viability of these cells, using a Prestoblue assay. We found that the GI50 for PTX (7 nM) was higher in this cell line than in HeLa cells, as expected [16] (Appendix A). When used in combination with PTX, Carba1 was able to synergistically affect cell viability, at high dose. Carba1 was not toxic for this cell line (Appendix A). These results indicate that Carba1 does not induce additional toxicity.

### 2.3. Cell-Cycle Progression Is Blocked at Mitosis by Carba1

A videomicroscopy analysis, using different doses of Carba1, showed that the compound impacted mitosis. As compared to DMSO, Carba1 (12 μM) induced a significant delay in the completion of metaphase and a slight increase of aberrant mitosis (Figure 3, Appendix A). When Carba1 was applied at a concentration of 25 μM, the majority of the cells stayed blocked in prometaphase (Figure 3 and Appendix A). We followed and quantified the fate of the cells treated with 25 μM Carba1 in a 20-h time lapse video (Appendix A) and noted that 61% of the mitotic cells eventually died during mitosis, 29% were still dividing abnormally, whereas only 10% underwent apparently normal mitosis (Appendix A). We thus concluded that a cytotoxic dose of Carba1 induced a very long duration of mitotic arrest, followed by mitotic catastrophe.

In accordance with the effect of Carba1 on mitosis, a flow cytometry analysis using propidium iodide staining indicated that a 15-h exposure to 25 μM Carba1 induced a dose-dependent cell-cycle arrest at the G2/M phase (Figure 4A). Prolonged exposure (24 and 48 h) led to a reduction of the number of cells blocked in the G2/M phase and to an increase of aneuploid cells, as assessed by the increased number of cells in sub G1 and of cells containing more than 4N DNA (Figure 4A).

### 2.4. Carba1 Increases PTX Effects on Cell Cycle and Mitosis

We similarly analyzed the effect of PTX, using time-lapse microscopy. PTX at a concentration of 1 nM induced a delay in chromosome congression during prometaphase and a moderate increase of aberrant mitosis (Appendix A). When treated with a cytotoxic concentration of PTX (5 nM) 80% of HeLa cells underwent aberrant mitosis followed by a mitotic slippage, as shown by a 12-h time-lapse video (Appendix A). We conducted a flow cytometry analysis to get further insight of the effect of 5 nM PTX on the fate of HeLa cells treated for longer times (15, 24, or 48 h). After a 15-h treatment, half of the cell population was blocked in the G2/M phase and nearly 20% of the cells were dead, as indicated by the increased proportion of cells in subG1. Then, the proportion of cells in G2/M gradually decreased, in parallel with an increased number of cells in subG1 and of multinucleated cells (Figure 4B). As the effects of such a cytotoxic concentration of PTX were different from those of a cytotoxic (25 μM) concentration of Carba1, we wondered which compound effect was predominant in the cytotoxicity of the combination of Carba1 and PTX (12 μM/1 nM). We thus compared the effects of this cytotoxic combination to the effects of Carba1 25 μM and PTX 5 nM administrated separately. As shown in Figure 4C, the combination of Carba1 and PTX (12 μM/1 nM) induced an arrest of the cell cycle almost superimposable to the arrest observed when cells are treated with PTX 5 nM. Moreover, the videomicroscopy analysis of the cells treated with this combination showed that cell death occurred after mitotic slippage (Appendix A). The similarity of the results obtained with the combination Carba1 and PTX (12 μM/1 nM) to those obtained with PTX 5 nM indicates that the overall effect of the combination results from an increase of the PTX effect induced by Carba1.

### 2.5. Carba1 Is a Microtubule-Destabilizing Agent

In an attempt to understand the Carba1 mechanism of action, we first analyzed its effect on cellular MTs, using immunofluorescence. Carba1 treatment (12–25 μM) did not visibly perturb the MT network in interphase cells, as compared to DMSO (control; Figure 5A). In mitosis, chromosome congression defects were visible in several mitotic cells of the 12 μM treated cell population. The occurrence of such defects was increased at a higher dose (25 μM) of Carba1 (Figure 5A). Such defects in chromosome congression are similar to those observed on cells treated by some inhibitors of kinases involved in the mitotic process such as Aurora B or Plk1 kinases [17]. Moreover, compounds structurally related to Carba1 often target protein kinases [18,19]. We therefore tested the ability of Carba1 to inhibit the activity of a panel of 64 protein kinases including kinases known to be involved in the regulation of the cytoskeleton and/or the cell cycle. We found that, when in vitro assayed at a 10 μM concentration, Carba1 did not show any selective inhibitory activity on the kinases tested (Appendix A). It is therefore unlikely that Carba1 is a direct inhibitor of these kinases.

The observed effects of Carba1 on the cellular MT network were reminiscent to those described for low doses of MT depolymerizing agents such as nocodazole or vinca alkaloids: a mitotic arrest with a similar aberrant chromosome organization, with no detectable effect on the total MT mass [20]. We thus wondered if Carba1 was, as nocodazole, able to directly impact MT assembly. The effect of a high dose (25 μM) of Carba1 on MT dynamic instability parameters was measured using time-lapse fluorescence microscopy on GFP-EB3 transfected cells (Appendix A). Carba1 reduced the MT growth rate as well as the MT growth, as indicated by the increase of the distance-based catastrophe frequency, and increased time spent in pause, indicating that Carba1 suppresses MT dynamics.

We therefore tested the Carba1 effect on in vitro tubulin assembly. As shown in Figure 5B, Carba1 was able to inhibit polymerization of pure tubulin in a dose-dependent manner. Increasing Carba1 doses induced a decrease in the rate of polymerization, as well as a delay in nucleation and a reduction in the total quantity of assembled MTs, attested by the scaling down of the level of assembly at equilibrium (Figure 5B). The concentration of Carba1, which inhibits 50% of tubulin (30 μM) assembly under these experimental conditions, was 6.9 μM.

We then looked for the binding site of Carba1 on tubulin. Among the four binding sites described for MT depolymerizing agents, the most common binding site is the colchicine site [1]. We checked whether Carba1 could compete with [^3^H]-colchicine for its tubulin binding-site (Figure 5C). Carba1 selectively inhibited colchicine binding to tubulin, indicating that it binds to tubulin at or near the colchicine site. However, it did not completely prevent the binding of [^3^H]-colchicine, suggesting that its affinity for this site is lower than that of colchicine.

In order to measure the binding constant of the compounds, a competition assay with 2-methoxy-5-(2,3,4-trimethoxyphenyl)-2,4,6-cycloheptatrien-1-one (MTC), an analogue of colchicine lacking the B ring that rapidly reaches an equilibrium (Kb = 4.7 × 10^5^ M^−1^, 25 °C [21]) in its binding reaction with tubulin, was designed. In the absence of tubulin the compound lacked fluorescence (Figure 5D) while in the presence of tubulin an emission maxima at 423 nm was observed upon excitation at 350 nm. As expected from its activity as an inhibitor of [^3^H]-colchicine binding to tubulin, Carba1 is able to displace MTC from the colchicine site, strongly supporting that Carba1 binds to the colchicine site of tubulin. The dissociation constant of Carba1, for the colchicine site is 3.03 ± 0.5 × 10^−6^ mol L^−1^ (Figure 5E). Altogether, these results show that Carba1 is a direct inhibitor of MT polymerization.

### 2.6. Carba1 Binding to Tubulin Enhances the Tubulin Binding Capacity of PTX and Its MT Stabilizing Activity

Recently, high resolution imaging of MT dynamics in vitro has demonstrated that fluorescent PTX analogs strongly accumulate at MT ends that are transitioning from growth to depolymerization, and, therefore, low non-saturating concentrations of a MT depolymerizing agent such as vinblastine, which enhances catastrophes, promote taxane binding to growing MT tips [13]. Such a mechanism could explain the observed synergy between Carba1 and PTX. To test if this mechanism is at work with Carba1, we first determined the Carba1 concentration able to induce catastrophes. We used a Total Internal Reflection Fluorescence microscopy (TIRF)-based assay, in which microtubules are grown from seeds stabilized with GMPCPP in the presence of microtubule plus end marker EB3 (Figure 6) [22]. Similar to our previous findings with other MT-destabilizing agents [22], we found that 10 μM Carba1 reduced MT growth rate and induced a two-fold increase of the catastrophe frequency (Figure 6A,D,E). Catastrophe frequency increased even further when Carba1 was combined with 100 nM PTX (Figure 6B,D,E), and moreover, we observed formation of sites where MTs were repeatedly rescued, resulting in frequent switching between MT growth and shortening (Figure 6B). As described recently, appearance of such “stable rescue sites” points to formation of “hotspots” of enhanced taxane accumulation at MT ends transitioning to catastrophe [13]. To confirm this idea, we visualized taxane binding to MTs by using a fluorescent PTX derivative Fchitax-3. As expected, the combination of Carba1 with 100 nM Fchitax-3 induced an increase of catastrophe frequency similar to the increase observed when Carba1 was combined with PTX (Figure 6C,E). We found that Carba1 indeed increased the frequency of accumulations of Fchitax-3, very similar to what we observed previously with vinblastine [13] (Figure 6C,F). This result strongly suggests that the two drugs act synergistically because Carba1-induced catastrophe induction leads to formation of “hotspots” of enhanced binding of PTX at growing MT ends during their transition to catastrophe. The observed effect is thus not due to the binding of Carba1 and PTX to the same tubulin dimer but rather due to the changes in MT lattice conformation which are induced by Carba1 and promote local accumulation of PTX.

### 2.7. Carba1 and PTX Act Synergistically to Reduce Tumor Growth In Vivo

Could the synergy between Carba1 and PTX, which we observed both at the level of individual MTs and at the cellular level, be translated into a therapeutic anti-cancer effect? To address this question, we compared the effects on tumor growth of Carba1 and PTX administrated separately to the effect of administration of Carba1 in combination with PTX in two animal models with different PTX sensitivity levels. We first evaluated the effect of the combination on the allogeneic 4T1-rvLuc2 mouse mammary carcinoma model. This model is known to be poorly sensitive to PTX and we hypothesized that the co-administration of Carba1 could enhance PTX sensitivity. Before starting the experiments on animals, we evaluated if Carba1 also exerts a synergy with PTX on the 4T1 cell line in vitro (Appendix A). The GI50 of PTX alone on these cells was 90 nM and was decreased by 1.8 times in the presence of Carba1 at 12 μM and by nine times in combination with Carba1 at 25 μM.

Tumor allografts were then established by implantation of 4T1 tumor cells into the mammary fat pad of nude mice. The mice were randomly assigned to five treatment groups: vehicle, Carba1 30 mg/kg, PTX 2 mg/kg, Carba1 30 mg/kg plus PTX 2 mg/kg, and PTX 8 mg/kg and the drugs were then daily injected intraperitoneally.

From 14 days of treatment, although the body weight of the animals was not significantly altered (Appendix A), some of the mice treated with PTX at 8 mg/kg (three out of eight) died or presented signs of suffering (prostration and abdominal distension) that require their killing for ethical reasons. Autopsy revealed that intestinal occlusion was the cause of those trouble and deaths. Interestingly, no death or signs of suffering were observed in the Carba1 and Carba1 plus PTX 2 mg/kg groups.

Tumor growth is presented in Figure 7A, and shows that whereas Carba1 30 mg/kg has no effect when administrated alone, PTX (2 mg/kg), PTX (8 mg/kg) as well as the combination Carba1 30 mg/kg plus PTX 2 mg/kg induced a significant decrease of the tumor volumes, when compared to vehicle-treated animals. However, although we observed a greater reduction in tumor growth after the Carba1/PTX combination treatment than after the exposure to PTX alone, this difference remains non-significant. We also analyzed the tumor cell viability using in vivo bioluminescence imaging with similar results (Appendix A).

Although these results suggest that Carba1 can synergize with PTX, neither the combination of Carba1 (30 mg/kg) + PTX (2 mg/kg) nor the highest dose of PTX alone (8 mg/kg daily administration) could break the tumor growth curve.

We thus concentrated our efforts on the fully PTX-sensitive HeLa tumor model whose response to high-dose PTX treatment is complete but concomitant to adverse effects that would be desirable to reduce without losing therapeutic effectiveness.

In a first series of experiments, we analyzed the effect of increasing doses of PTX or Carba1 when administered alone. To that aim, mice bearing already well-established tumors (>250 mm^3^) received intravenous (i.v.) injections of PTX (from 2 or 8 mg/kg), every 2 days during 10 days (Figure 7B). In the same experiment, we analyzed the effect of Carba1 (from 15 to 60 mg/kg, i.v.) injected with the same schedule (Figure 7C). We found that PTX, when administered at 8 mg/kg, induced a significant reduction of tumor size (Figure 7B). Carba1 did not induce a significant effect on tumor size whatever the dose injected, although a tendency towards smaller tumors appears with increasing Carba1 concentrations (Figure 7C). The results confirmed the anti-tumor effect of high PTX concentrations in this model. They also indicate that Carba1, when applied alone, has no significant anti-tumor activity, even at high concentrations. As shown in Appendix A the weight of PTX or Carba1 treated animals and vehicle-treated animals were not significantly different. Moreover, the animals did not show any sign of discomfort, suggesting a good tolerance of the treatments.

We then conducted a study of the effect on tumor size of a low PTX dose (3 mg/kg) in combination with Carba1 (60 mg/kg). No animal death and no modification of body weight were observed throughout the study (Appendix A). In this experiment neither PTX (3 mg/kg), nor Carba1 (60 mg/kg) has an effect on tumor size (Figure 7D). As shown in Figure 7D, while the size of tumors still increases when each compound is administered separately, an arrest of tumor growth is observed with the combination of PTX and Carba1. These results indicate that the observed synergy between PTX and Carba1 in vitro also occurs in vivo, leading to an enhanced therapeutic efficacy at a low-dose of PTX treatments.

## 3. Discussion

Our initial aim was to discover an agent that would allow lowering the dose of PTX while obtaining the same anti-tumor efficacy as the currently used therapeutic dose of PTX. We thus screened a chemical library to detect compounds able to sensitize cells to a low, non-toxic dose of PTX. The test we used was a cytotoxicity test, therefore probing all vital cell functions. Whereas such a whole cell-based assay screens molecules having multiple potential targets and allows the biology to dictate the best targets [23], it may not be insignificant to have selected Carba1, an agent that targets tubulin and impairs MT dynamics. Indeed, this indicates that the most sensible target, in this specific context, is tubulin.

Recently, a series of carbazole-based MT targeting agents has been reported [24]. These acyl-substituted derivatives, conceived as analogs of nocodazole, represent other examples of carbazole scaffolds able to interact with the colchicine site of tubulin. Unlike Carba1, they exert potent killing activities in human glioblastoma cells. A possible explanation for the difference in cytotoxicity observed between Carba1 and the compounds described by Diaz et al. [24] may reside in a lower affinity of Carba1 for the colchicine binding site of the tubulin dimer. Indeed, we found that the affinity of Carba1 for the colchicine site is not very high, in the micromolar range. This difference in affinity could be due to the presence of the pyrol ring on Carba1, which could generate steric hindrance and decrease the affinity of the compound for the colchicine site. In addition, the work of Diaz et al. emphasizes the importance of substitutions at the level of the nitrogen atom of the carbazole moiety. Such substitutions are absent from Carba1 and thus may impact binding of the compound to the colchicine site.

The Carba1 scaffold is a versatile one and we are currently synthesizing modified analogs for medicinal chemistry optimization.

The PTX binding site at the interior of the MT has been characterized at the atomic level: PTX binds to a pocket in β-tubulin that faces the MT lumen and is near the lateral interface between protofilaments (for review see [1]). The binding of PTX results in the expansion of the taxane binding pocket [25] of the tubulin dimer. Moreover, PTX binding inhibits, in the protofilament, the compaction at the longitudinal interdimer interface, induced by GTP hydrolysis [26]. This allosteric mechanism would strengthen the longitudinal tubulin contacts leading to a stabilization of the MTs [1]. In this context, it is conceptually counterintuitive that an agent that depolymerizes MTs acts in synergy with PTX, an agent that stabilizes them.

A possibility is that the binding of Carba1 to the tubulin dimer modifies its affinity for PTX. However, although it has been shown that the covalent occupancy of the taxane site can affect the structure of the colchicine site [27], the reverse has not yet been described. Moreover, in cells, due to the low affinity of Carba1 for tubulin and the nanomolar concentration of PTX that was used, it can be assumed that the probability that a single tubulin dimer has both a molecule of Carba1 and another of PTX bound is very low. Thus, an allosteric effect at the level of the tubulin dimer, due to such a simultaneous binding, cannot be responsible for synergistic cytotoxicity.

Another possibility is that the binding of Carba1 can induce conformational changes of the growing MT ends that can facilitate the subsequent binding of PTX to the MT lattice. Recently, using TIRF analysis, it has been shown that non-saturating doses of vinblastine induce a switch to catastrophe and convert the MT plus end to a state that allows more efficient taxane accumulation [13]. Indeed, we conducted the same type of experiment, replacing vinblastine with Carba1 and observed an increase in the rate of catastrophes associated with more frequent formation of accumulation “hotspots” of fluorescent taxane. Moreover, we observed a synergistic effect of Carba1 and PTX on MT growth inhibition in vitro. Although the underlying structural mechanism is yet unknown, it is highly probable that Carba1 acts similarly to vinblastine to facilitate PTX accumulation. Our experimental in vitro data thus strongly suggest that Carba1 acts by modifying the growing tip of microtubules, favoring the accumulation of PTX.

It is known that PTX accumulates intracellularly [4], reaching intra-tumor concentrations that are higher in the tumors than in the plasma [7]. It is thus remarkable that the synergistic effect is observed not only at the MT level, but also at the cellular level, as well as when both drugs are applied systemically in animals to exert their anti-tumor action. Although the most probable hypothesis is that the same molecular mechanism is at work in these different contexts, it cannot be excluded that Carba1 has other target(s). For instance, Carba1 could also inhibit a drug export pump and further facilitate PTX accumulation.

We observed a different anti-tumor response in the two models used. In one case (allogeneic orthotopic 4T1 cell transplant model) the difference in tumor size observed between PTX alone and Carba1/PTX combination therapy was not significant. In the other case (xenograft of HeLa cells), a synergistic effect is clearly demonstrated.

We assume that these different tumor responses reflect the sensitivity of these cells to PTX in vitro (PTX GI50 is 1.5 nM in HeLa cells and 90 nM in 4T1 cells). More trivially, differences in response between the two models may also result from the protocols used, particularly with respect to the injection route—i.p. (4T1) versus i.v. (HeLa)—and the frequency of injections—every day (4T1) versus every other day (HeLa). Nor can we rule out the possibility that the combination is more effective on certain tumor types. A thorough study of the effect of the combination on different cell lines, such as the one performed for Carba 1 alone in the NCI60 screen, will probably answer this question.

These last years, attention has been directed to the combined use of therapeutic agents to target critical cellular pathways involved in carcinogenesis. Accordingly, several small molecules have been reported to synergize with PTX to kill cancer cells. For instance, combined administration of the src inhibitor dasatinib and PTX has a synergistic antiproliferative activity on different cancer cell lines [28,29]. Similarly, aberrations in the PI3K/AKT/mTOR pathway are commonly described in aggressive cancers [30] and inhibitors of this pathway can improve outcomes in some patients when combined with paclitaxel [30,31,32]. We cannot completely exclude that Carba1 exerts its synergistic effect by additionally targeting a pathway involved in cell survival. However, it is less probable that it targets a survival pathway involving some kinases as we have tested Carba1 potential inhibitory activity on a panel of 64 kinases, including src and mTOR, and found that Carba1 did not exhibit any selective activity on the assayed kinases.

As the combined administration of Carba1 and a low dose of PTX can have an anti-tumor effect, one could imagine that this combination should reduce the unwanted side effects observed with high doses of PTX. This has to be tested. For instance, the effect of the combination should be compared to the PTX effect on the kinesin-based anterograde transport, since perturbation upon PTX treatment is thought to be part of the mechanism involved in peripheral neuropathy [12]. Given the mode of action we have described, with Carba1 facilitating the accumulation of PTX in MTs, we can bet that the combination should diminish MT-independent adverse events. Interestingly, it has been shown that a compound sharing the same carbazole scaffold protects against PTX-induced peripheral neuropathy [33].

Anti-cancer strategies based on the concomitant administration of taxanes and depolymerizing agents such as vinorelbine have been reported [34,35,36]. However, these approaches used high doses of each of these drugs. Our results suggest that good therapeutic efficacy could be achieved with the combined administration of each of these agents at low doses, which could improve patient comfort.

In addition, reducing the doses of taxane administered to patients could delay the onset of acquired resistances. This work thus paves the way to new therapeutic perspectives that are easy to implement.

## 4. Materials and Methods

### 4.1. Chemical Reagents and Cells

The chemical library that was used for the initial screening is part of the collection assembled by the CERMN (Centre d’Études et de Recherche sur le Médicament de Normandie, University of Caen, Caen, France). This collection is composed of original compounds (>19,000), synthesized within the frame of the numerous drug design programs the unit has developed for 40 years. These compounds are mainly small molecules, often including heteroelements and heterocycles, they basically obey to the Lipinski’s rules of five and are considered as druggable. The 8000 compounds screened were selected from this chemical library using a clustering method in order to adapt the size of the panel to the capacity of the screening, while respecting the structural diversity of the collection.

Carba1 was re-synthesized at the CERMN and supplied as a powder. It was dissolved in anhydrous dimethyl sulfoxide (DMSO, Sigma-Aldrich, #D4540, St Quentin Fallavier, France) and stored at −20 °C as 10 mM stock solution. Paclitaxel (PTX) was purchased from Sigma (#T7402) and was dissolved in DMSO and stored at −20 °C as 1 mM stock solution.

The human HeLa and RPE-1 cell lines and the murine 4T1 cell line were obtained from the American Type Culture Collection (ATCC, Gaithersburg, MD, USA), routinely tested and authenticated by the ATCC. HeLa Kyoto cells expressing EGFP-alpha-tubulin and H2B-mcherry were from Cell Lines Service, #300670. HeLa cells and 4T1 cells were grown in RPMI 1640 medium (Gibco, Invitrogen, Carlsbad, CA, USA) and RPE-1 cells were grown in DMEM (Gibco, Invitrogen), supplemented with 1% penicillin/streptomycin and 10% Fetal Bovine Serum, and maintained in a humid incubator at 37 °C in 5% CO_2_.

### 4.2. Analysis of Cell Viability Using MTT (Screening of the Chemical Library)

The assay was performed in 96-well microplates. Cells were seeded at a density of 2500 cells per well and allowed to adhere for 24 h before being treated for 48 h with either DMSO (0.1% final concentration) or compounds at 5 μM, with or without 1 nM PTX. Viability was evaluated with a 3-(4,5-dimethylthiazol-2-yl)-2,5-diphenyl-tetrazolium bromide (MTT) colorimetric assay (Sigma, #M5655).

### 4.3. Analysis of Cell Viability Using Prestoblue Assay

Cell viability was analyzed using the colorimetric Prestoblue assay (Invitrogen, #A13262). Cells were seeded in 96-well microplates (Greiner, #655077, Courtaboeuf, France) at a density of 2500 cells per well and allowed to adhere for 24 h before being treated for 72 h with either DMSO (0.1 % final concentration) or drugs at indicated concentrations. After a 72-h treatment, 11 µL Prestoblue was added to each well and cells were incubated for another 45 min. The absorbance of each well was measured using a FLUOstar Optima microplate reader (Excitation, 544 nm; Emission, 580 nm, BMG Labtech, Champigny/Marne, France).

### 4.4. Apoptosis Assay

The apoptosis assay was performed with FITC Annexin V Apoptosis Detection Kit I (BD Biosciences, #556547, San Jose, CA, USA) using flow cytometry and analyzed by FCS express software.

### 4.5. Drug Combination Analysis

The Chou–Talalay analysis on the basis of dose–response curves was used to evaluate the synergism, additivity, and antagonism of the combination drug treatment. Combination index (CI) values were calculated using the CompuSyn software (www.combosyn.com) which uses the equation:CI = CA,×/SA,× + CB,×/SB,×
where CA,× and CB,× are the concentrations of drug A and drug B in the combination to produce a certain effect ×. SA,× and SB,× are the concentrations of drug A and drug B used as a single agent to produce that same effect. CompuSyn also generates a plot of CI values at different fraction affected (Fa) levels referred to as Fa-CI plot, which are widely used to interpret drug combination effects [37]. A CI value of <0.1 indicates very strong synergism, 0.1–0.3 strong synergism, 0.3–0.7 synergism, 0.7–0.9 moderate to slight synergism, 1 additivity, 1.1–1.45 slight to moderate antagonism, 1.45–3.3 antagonism, and >3.3 strong to very strong antagonism.

### 4.6. Cell Cycle Analysis

Cells were harvested and washed by centrifugation in PBS. Then, 10^5^ cells were fixed in 1 mL of 70% ethanol at 4 °C overnight. Following two washes with PBS the cells were incubated with 50 µg/mL propidium iodide and 0.2 mg/mL RNase A (Sigma, #10109142001)/PBS for 30 min at 37 °C before analysis. The percentage of cells in the specific cell-cycle phases (G0, G1, S, G2, and M) was determined using an Accuri C6 flow cytometer (Becton Dickinson, San Jose, CA, USA).

### 4.7. Analysis of Carba1 Effect on Kinases

The analysis of Carba1 effect was performed on a panel of 64 recombinant protein kinases. The assays were performed at 10 μM ATP in the presence of 10 μM Carba1 using the Upstate Kinase profiler panel service (Millipore, Molsheim, France). Inhibition, expressed as the percent of activity determined in the absence of inhibitor, was calculated from the residual activity measured in the presence of 10 μM Carba1.

### 4.8. Immunofluorescence Microscopy and Live Cell Imaging

HeLa cells at a density of 20,000 cells were grown for 48 h on glass coverslips placed in a 24-well microplate. When cells reached 70% confluence the medium was replaced with a fresh one supplemented with Carba1. After a 5-h exposure to Carba1, cells were fixed and permeabilized with −20 °C absolute methanol for 6 min. After washing and saturation with 3% BSA (Bovine Serum Albumin; Sigma, #A7906)/PBS (Phosphate Buffered Saline; Dutscher, #L0615-500, Brumath, France), cells were incubated for 45 min at room temperature (RT) with anti-alpha-tubulin antibody (clone α3A1, 1:4000), produced by L. Lafanechère [38]. Cells were washed twice again and subsequently incubated with Alexa 488 conjugated anti-mouse antibody (1:1000, Jackson immunoresearch, #115-545-4637, Cambridgeshire, UK) for 30 min at RT. DNA was stained with 20 µM Hoechst 33342 (Sigma, #23491-52-3) and coverslips were mounted on glass slides with Mowiol 4-88 (Calbiochem- Sigma-Aldrich, #475904, St Quentin Fallavier, France). Images were captured with a Zeiss AxioimagerM2 microscope equipped with the acquisition software AxioVision (Marly le Roi, France) and analyzed using the Fiji software (imagej.net). For live-cell imaging, HeLa Kyoto cells expressing EGFP-alpha-tubulin and H2B-mcherry were seeded on 2-well glass-slides (Ibidi, #80297, Gräfelfing, Germany) at a density of 7000 cells per well and allowed to grow for 24 h prior to imaging. After treatment, the slide was placed on a 37 °C heated stage, at 5% CO_2_, and images were acquired every 2.5 min by a spinning disk confocal laser microscope (Andromeda iMIC, TILL I.D. GmbH, Martinsried, Germany) equipped with a Plan-Apochromat 20×/0.75 WD610 objective and an EMCCD camera (iXon 897, Oxford Instrument, Belfast, UK). For each time point, a stack of seven planes (thickness: 1 µm) was recorded. Acquisition (LA), off-line analysis (OA) and Fiji software programs were used.

### 4.9. Transfection of GFP-EB3

To label MT plus ends, GFP-EB3 plasmids were used because EB3 has a strong binding affinity to MT plus ends. Cell transfection was performed using electroporation (AMAXA^®^, Köln, Germany). In total, 2 µg of purified plasmid DNA were used for each transfection reaction.

### 4.10. Fluorescence Time-Lapse Videomicroscopy of MT Plus Ends

Live imaging of MT plus ends was performed as described in Honoré et al. [39], on transiently GFP-EB3 transfected-HeLa cells by using an inverted fluorescence microscope (ZEISS Axiovert 200M with a 63× objective, Zeiss, Marly le Roi, France); Time-lapse acquisition was performed with a COOLSNAP HQ (Roper Scientific, Ottobrunn, Germany), driven by Metamorph software (Universal Imaging Corp., Molecular Devices, San Jose, CA, USA). Image acquisition was performed at a temperature of 37 ± 1 °C/5% CO_2_.

Data are from three independent experiments. For each experiment, six MTs/cell in six cells per condition were analyzed.

### 4.11. Dynamic Instability Parameter Analysis

The dynamic instability parameter analysis was performed by tracking MT plus ends over time using the imageJ software (imagej.net). The methods of calculation were as described in Honoré et al. [39].

### 4.12. Tubulin Polymerization Assay

Tubulin was prepared from bovine brain as previously described [40]. Tubulin polymerization assays were carried out at 37 °C in BRB80 buffer (80 mM Pipes, 0.5 mM MgCl_2_, 2 mM EGTA, 0.1 mM EDTA, pH 6.8 with KOH) by mixing 7 µM of pure tubulin, 1 mM GTP, 5 mM MgCl_2_, and indicated concentrations of drugs (0.2% DMSO, final concentration) in a final volume of 100 µL. The time course of the self-assembly activity of tubulin was monitored as turbidity at 350 nm, 37 °C, during 30 min, using a spectrophotometer (ThermoScientific, Evolution 201, Waltham, MA, USA).

### 4.13. [^3^H]-Colchicine Tubulin-Binding Assay

The tubulin was prepared from bovine brain as previously described [40]. Pure tubulin (3 µM final concentration) in cold BRB80 buffer was mixed at 4 °C with a mix of [^3^H]-colchicine (82.6 Ci/mmol, Perkin-Elmer, #NET189250UC, 50 nM final concentration, Courtaboeuf, France) and the competitor Carba1 (100 µM final concentration) in a final volume of 200 µL. Following a 30-min incubation at 30 °C, the samples were deposited onto 50 µL of presedimented DEAE Sephadex A25 in BRB80 buffer. All subsequent steps were carried out at 4 °C. Samples were incubated for 10 min with continuous shaking to ensure quantitative binding of tubulin to the gel. Following centrifugation (2400 *g*, 4 min), supernatants were discarded and the pellets containing the bound molecule-tubulin complexes were washed four times with 1 mL of BRB80 buffer. Pellets were incubated for 10 min with 500 µL of ethanol to solubilize the tubulin-bound tritiated colchicine and 400 µL aliquots of the ethanol solutions were transferred to 5 mL of Ultima Gold scintillant (Perkin-Elmer) for determination of radioactivity.

### 4.14. Determination of the Binding Constant of Carba1 on Tubulin Using a Competition Assay

Calf brain tubulin was purified as described [41]. 2-Methoxy-5-(2,3,4-trimethoxyphenyl)-2,4,6-cycloheptatrien-1-one (MTC) [42] was a kind gift of Prof. T. J. Fitzgerald (School of Pharmacy, Florida A & M University, Tallahassee, FL, USA). The compounds were diluted in 99.8% DMSO-d6 (Merck, Darmstadt, Germany) to a final concentration of 10 mM and stored at −80 °C.

Competition of the compound with MTC was tested by the change in the intensity of fluorescence of MTC upon binding to tubulin. The fluorescence emission spectra (excitation at 350 nm) of 10 μM tubulin and 10 μM MTC in 10 mM sodium phosphate, 0.1 mM GTP, pH 7.0, were measured in the presence of different concentrations (0–20 μM) of the desired ligand with 5 nm excitation and emission slits using a Jobin-Ybon SPEX Fluoromax-2 (HORIBA, Ltd., Kyoto, Japan). The decrease in the intensity of the fluorescence in the presence of the competitor ligand indicated competition for the same binding site. The data were analyzed and the binding constants determined using Equigra V5.0 (available upon request to J.F. Diaz) as described in Díaz and Buey [43].

### 4.15. In Vitro MT Dynamics and Analysis of MT Dynamics Parameters

In vitro assay for MT growth dynamics and analysis of MT dynamic parameters in the presence of tubulin (Cytoskeleton Inc., Denver, CO, USA) and EB3 with Carba1 and Fchitax-3 was performed as described previously [13]. For statistical analysis, graphs were plotted in GraphPad Prism 7 (San Diego, CA, USA) and statistical analysis was done using non-parametric Mann–Whitney U-test.

### 4.16. Tumor Xenografts in Mice

All animal studies were performed in accordance with the institutional guidelines of the European Community (EU Directive 2010/63/EU) for the use of experimental animals and were authorized by the French Ministry of Higher Education and Research under the reference: apafis#8854-2017031314338357 v1.

In a first exploratory experiment, the effects of PTX and/or Carba1 were evaluated on the allogeneic 4T1-rvLuc2 orthotopic mammary carcinoma model. Five-week-old female NMRI nude mice (Janvier Labs, Le Genest-Saint Isle, France) were anesthetized (4% isoflurane/air for anesthesia induction and 1.5% thereafter) and 20,000 4T1 cells in 1× PBS were implanted into the mammary fat pad. Mice were then randomized in five groups of eight mice each and drugs were administered intraperitoneally every day. A first group received the vehicle (14% DMSO, 14% Tween 80, and 72% PBS). Two groups received PTX at different doses (2 and 8 mg/kg) while one other group received Carba1 at 30 mg/kg. A last group received a combination of Carba1 (30 mg/kg) and PTX (2 mg/kg). Tumor size was measured every 2 days, using a caliper, and the tumor volume was calculated as follows: length × (width)^2^ × 0.4. At the end of experiment, 300 µL of D-luciferin (Promega, 10 mg/mL in PBS, Madison, WI, USA) was injected intraperitoneally and the tumor cell viability was evaluated using in vivo bioluminescence imaging (IVIS Kinetic, Perkin-Elmer, Courtaboeuf, France). Living Image software (Perkin-Elmer) was used to analyze the results. Kruskal–Wallis test was used to compare the effects of the treatments on tumor size.

In a second series of experiment, the effects of PTX and/or Carba1 were evaluated on a tumor model based on HeLa cells transplantation. First, the effects of PTX or Carba1 when administrated alone were evaluated. To that aim, 5-week-old female NMRI nude mice (Janvier Labs, Le Genest-Saint Isle, France) were anesthetized (4% isoflurane/air for anesthesia induction and 1.5% thereafter) and were injected subcutaneously in the flank with 10^7^ exponentially dividing HeLa cells in 1 × PBS. Tumor size was measured three times a week using a caliper. When tumors reached a volume of about 250 mm^3^ i.e., 9 days after cell injection, mice were randomized in seven groups of six mice each and drugs were injected intravenously every 2 days. A first group received the vehicle (14% DMSO, 14% Tween 80, and 72% PBS). Three groups received PTX at different doses (2, 4, and 8 mg/kg) while three other groups received Carba1 at different doses (15, 30, and 60 mg/kg). Statistical comparison between mice groups were determined using two-way ANOVA.

Then, the effect of a combination of PTX- Carba1 was evaluated, and compared to the effect of the compounds alone. To that aim, 5-week-old female NMRI nude mice were injected subcutaneously with 10^7^ exponentially dividing HeLa cells into the right flank. When tumors reached a volume of about 200 mm^3^ i.e., 9 days after cell injection, mice were randomized in four groups of eight mice each and drugs were injected intravenously every 2 days. The first group received PTX at 3 mg/kg, the second group Carba1 at 60 mg/kg, the third group received a combination of Carba1 (60 mg/kg) and PTX (3 mg/kg), and the fourth group received the vehicle (14% DMSO, 14% Tween 80, and 72% PBS). For statistical analysis, we verified the normality of the data using a Shapiro test and the homogeneity of the variances using a Bartlett test. We then used a Student’s *t*-test to compare the different groups to the control (i.e., vehicle). We found that the *p* values of the comparisons of the group treated with PTX (3 mg/kg) and of the group treated with Carba1 (60 mg/kg) with the vehicle were, respectively, 0.5355 and 0.5139. These *p* values are greater than 0.05, indicating that these groups are not different from the vehicle group at the risk beta calculated greater than 90% (power of the test lower than 10%). On the contrary, the comparison of the group treated with the combination to the control group gave a *p* value of 0.00015, indicating that the combination has an effect on the tumor size, at the risk alpha of 5%.

## 5. Conclusions

A new mechanism favoring paclitaxel binding to dynamic microtubules can be transposed to in vivo mouse cancer treatments, paving the way for new therapeutic strategies combining low doses of microtubule targeting agents with opposite mechanisms of action.

## Figures and Tables

**Figure 1 cancers-12-02196-f001:**
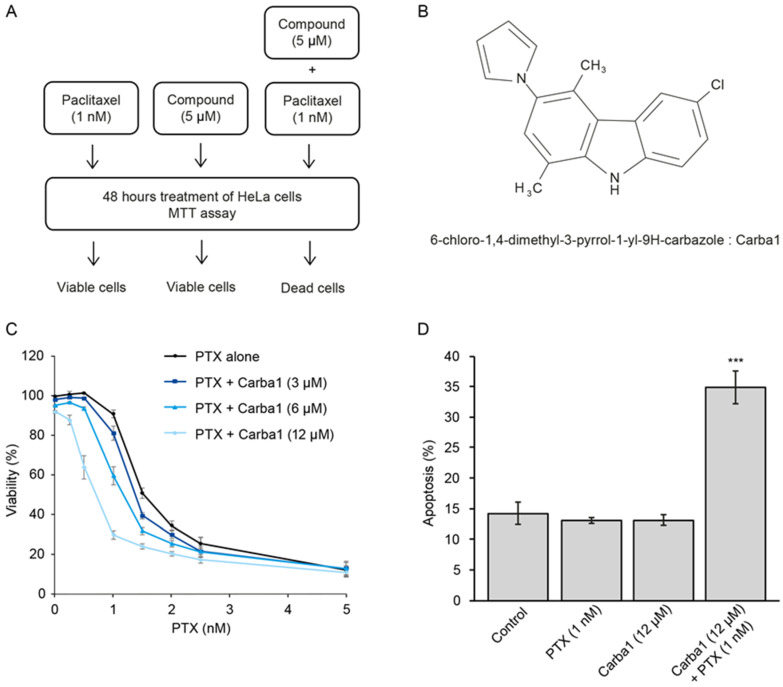
Selection of a compound that sensitizes cells to Paclitaxel (PTX). (**A**) Schematic illustration of the concept used to screen a chemical library for compounds that sensitize cells to PTX. Treatment of cells with compounds of the library alone (5 μM) or PTX (1 nM) alone has no effect on cell viability, compounds (5 μM) that have no effect when applied alone but induced cell death when applied in combination with PTX (1 nM) were selected. (**B**) Chemical structure of Carba1. (**C**) Effect of Carba1/PTX combinations on the viability of HeLa cells. Cells were incubated for 72 h with the indicated combinations of Carba1/PTX. The percentage of viable cells was calculated following a Prestoblue assay. Data are presented as mean ± SEM of three independent experiments. Note that the effect of the different doses of Carba1 alone on cell viability can be seen at the x-coordinate = 0, corresponding to 0 nM PTX. (**D**) Effect of Carba1 (12 μM), PTX (1 nM), and the combination of Carba1 and PTX (12 μM/1 nM) on HeLa cell apoptosis. HeLa cells, treated with the indicated concentrations of drugs, were stained with propidium iodide and annexin V and analyzed by flow cytometry. Results are expressed as mean ± SEM of three separate experiments. The significance was determined by a Student’s *t*-test (*** *p* < 0.001, compared to the control).

**Figure 2 cancers-12-02196-f002:**
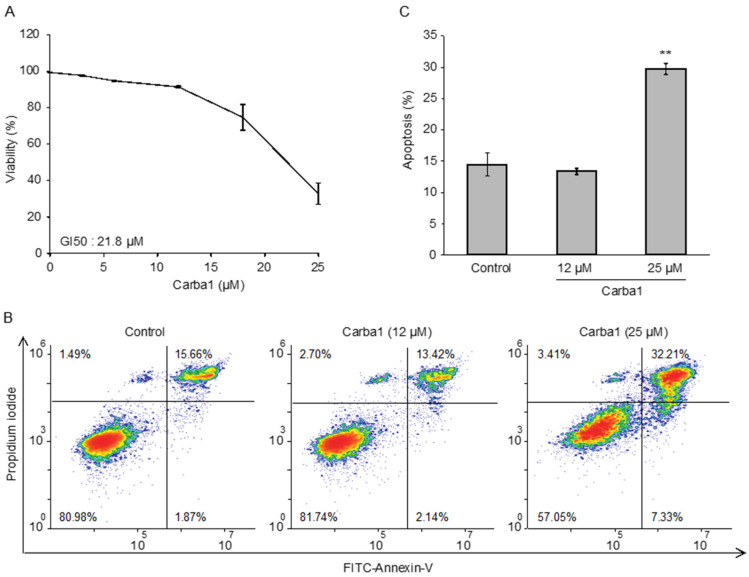
Carba1 has a low cytotoxicity. (**A**) Effect of Carba1 on HeLa cell viability. Cells were incubated for 72 h with increasing concentrations of Carba1. The percentage of viable cells was calculated following the Prestoblue assay. The results are expressed as mean ± SEM of three separate experiments. (**B**) Effect of Carba1 on HeLa cells apoptosis. HeLa cells, treated with the indicated concentrations of Carba1 for 72 h, were stained with propidium iodide and FITC-annexin V and analyzed by flow cytometry. Apoptotic cells are observed in the upper right part of the graphs. (**C**) Results for apoptotic cell death (as shown in Figure 2B) are expressed as mean ± SEM of three separate experiments. The significance was determined by a Student’s *t*-test (** *p* < 0.01, compared to the control).

**Figure 3 cancers-12-02196-f003:**
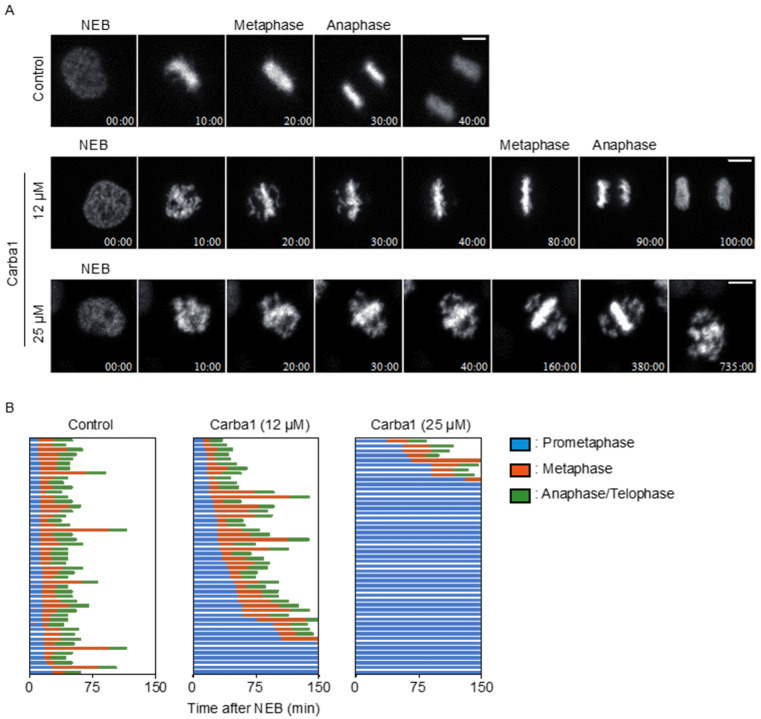
Carba1 induces a mitotic arrest. (**A**) Representative images, selected from Appendix A of HeLa Kyoto cells treated with DMSO (control) and the indicated concentrations of Carba1. Bar = 10 μm. (**B**) Analysis of the duration of mitosis in HeLa Kyoto cells treated with DMSO (control) or with different doses of Carba1, as indicated. Duration of prometaphase (from nuclear envelope breakdown (NEBD) to chromosome alignment; blue), metaphase (from chromosome alignment to anaphase onset; orange) and anaphase/telophase (from anaphase onset to chromosome decondensation; green) were analyzed from Appendix A. The data represent 50 cells for each treatment.

**Figure 4 cancers-12-02196-f004:**
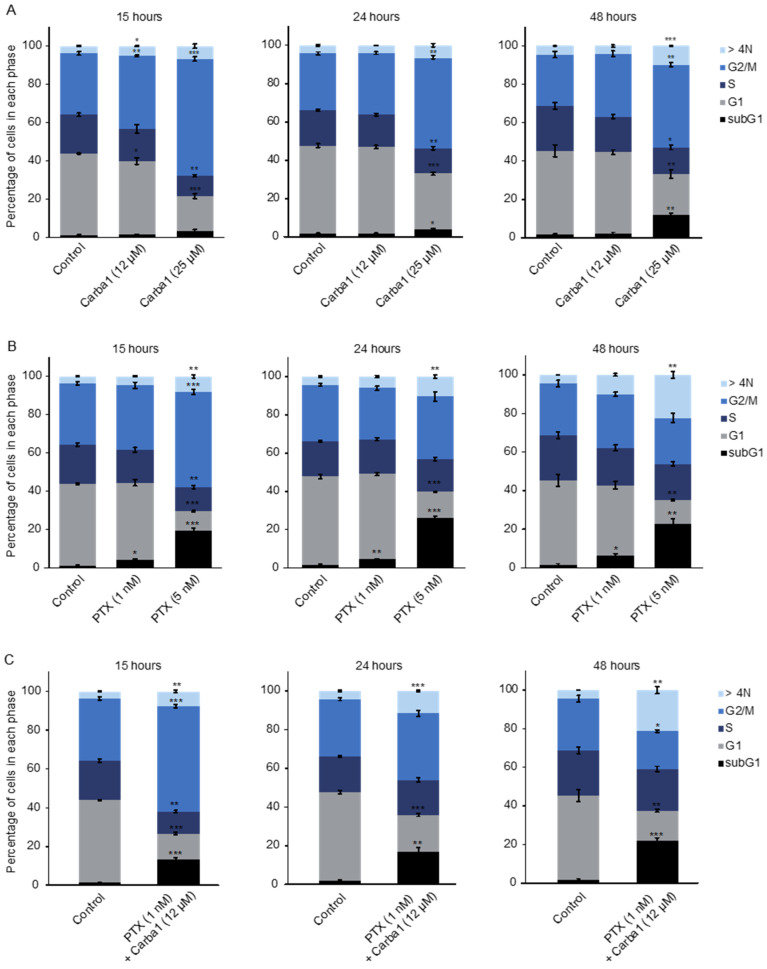
Carba1 increases PTX effects on cell cycle. (**A**) HeLa cells treated with the indicated concentrations of Carba1 for 15, 24, and 48 h, were fixed with 70% ethanol, stained with propidium iodide, and analyzed by flow cytometry. (**B**) HeLa cells treated with the indicated concentrations of PTX for 15, 24, and 48 h, were fixed with 70% ethanol, stained with propidium iodide, and analyzed by flow cytometry. (**C**) HeLa cells treated with the combination of Carba1 and PTX (12 μM/1 nM) for 15, 24, and 48 h, were fixed with 70% ethanol, stained with propidium iodide, and analyzed by flow cytometry. The results are expressed as mean ± SEM of three separate experiments. The significance was determined by a Student’s *t*-test (* *p* < 0.05, ** *p* < 0.01, *** *p* < 0.001, compared to the control).

**Figure 5 cancers-12-02196-f005:**
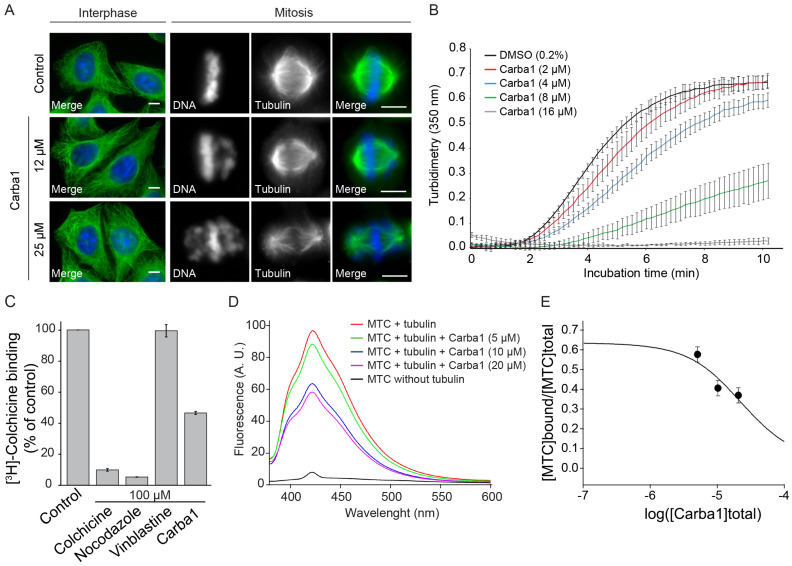
Carba1 is a microtubule-depolymerizing agent that competes with colchicine for tubulin binding. (**A**) Immunofluorescence analysis of the Carba1 effect on interphase and mitotic microtubules (MTs). MTs in interphase (left panels) or in mitosis (right panels) were stained using an anti-tubulin antibody, as described in the methods section. DNA was stained using Hoechst reagent. Bars = 10 μm. (**B**) Time course of tubulin polymerization at 37 °C in the presence of vehicle (DMSO, black line) and Carba1 at different concentrations (colored lines) as indicated, measured by turbidimetry at 350 nm. Purified tubulin: 30 µM in BRB80 buffer with 1 mM GTP. Each turbidity value represents the mean ± SEM from three independent experiments. (**C**) Effect of Carba1 on the binding of [^3^H]-colchicine to tubulin. Carba1 (100 µM) was used to compete with [^3^H]-colchicine (50 nM) as described in the methods section. Each value represents the mean ± SEM of three independent experiments. Colchicine and nocodazole were used as positive and vinblastine as negative control. (**D**) Displacement of 2-methoxy-5-(2,3,4-trimethoxyphenyl)-2,4,6-cycloheptatrien-1-one (MTC) from the colchicine site. Fluorescence emission spectra of 10 μM MTC and 10 μM tubulin in 10 mM phosphate-0.1 mM GTP buffer pH 7.0, in the absence or presence of increasing concentrations of Carba1. (**E**) Displacement isotherm at 25 °C of the fluorescent probe MTC (10 μM) bound to tubulin (10 μM) by Carba1 (black line and circles). The solid line is the best-fit value of the binding equilibrium constant of the competitors, assuming a one-to-one binding to the same site.

**Figure 6 cancers-12-02196-f006:**
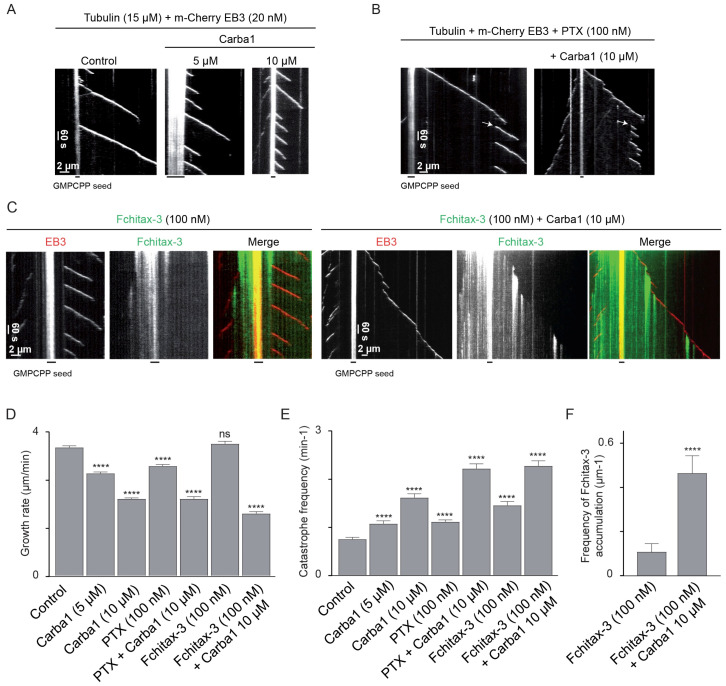
Carba1 binding to tubulin enhances the tubulin binding capacity of PTX. (**A**–**C**) Representative kymographs illustrating MT plus end growth in the presence of 15 µM tubulin and 20 nM m-Cherry EB3 (**A**) without (control) or with 5 and 10 µM Carba1; (**B**) PTX (100 nM) without or in combination with 10 µM Carba1. Arrows indicate repeatedly rescued sites; (**C**) Fchitax-3 (100 nM) without or with 10 µM Carba1. (**D**,**E**) Parameters of MT plus end dynamics determined using acquisitions as shown in (**A**–**C**) (three movies per condition, two independent experiments). The total number of analyzed growth events is 82 for 100 nM PTX, 72 for 100 nM PTX with 10 μM Carba1, and 100 for all other conditions. Each value represents the mean ± SD. Each condition was compared to the control condition using a Mann–Whitney analysis. (**F**) Quantification of Fchitax-3 accumulation frequencies per MT unit length in the presence of 15 μM tubulin and 20 nM mCherry-EB3 without (*n* = 11) or with 10 µM Carba1 (*n* = 13). Values represent the mean ± SD. Both conditions were compared using a Mann–Whitney analysis. *p* < 0.0001 (****), not significantly different (ns).

**Figure 7 cancers-12-02196-f007:**
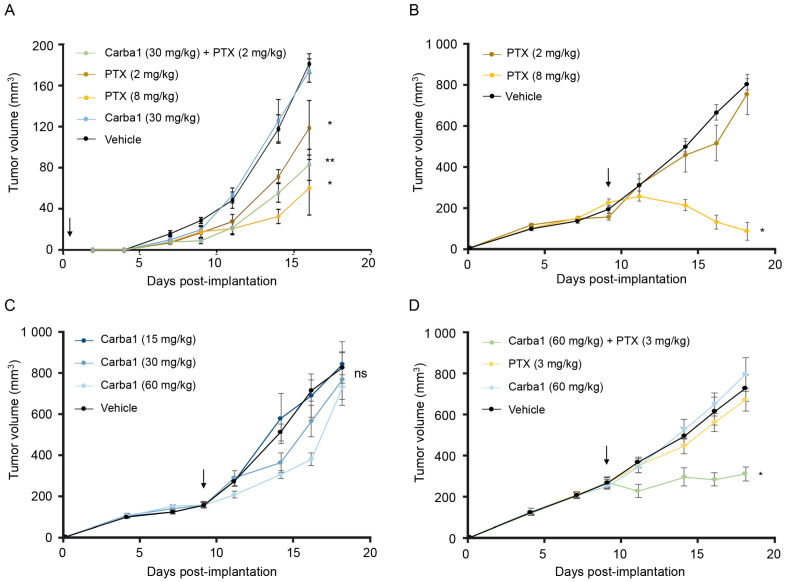
Carba1 acts synergistically with PTX to reduce tumor growth in vivo. (**A**) Exploratory analysis of the effect of PTX, Carba1, and their combination on the growth of 4T1 cells allografted in mice. Mice were treated with PTX (2 and 8 mg/kg), Carba1 (30 mg/kg), the combination of PTX (2 mg/kg) and Carba1 (30 mg/kg), or the vehicle. Tumor growth was monitored with a sliding caliper. Data represent a median. Error bars = MAD, * *p* < 0.05, ** *p* < 0.01 compared to vehicle (Kruskal–Wallis), *n* = 8 mice per group. (**B**) PTX inhibits the growth of HeLa cells xenografted in mice. When the tumors have reached a volume of about 200 mm^3^, mice were treated with PTX (2 and 8 mg/kg) or the vehicle. Tumor growth was monitored with a sliding caliper. Error bars = SEM, * *p* < 0.01 compared to vehicle (ANOVA), *n* = 6 mice per group. (**C**) Carba1 has no significant effect on the growth of HeLa cells xenografted in mice. When the tumors have reached a volume of about 200 mm^3^, mice were treated with Carba1 (15, 30, and 60 mg/kg) or the vehicle. Tumor growth was monitored with a sliding caliper. Error bars = SEM, ns = non-significant (ANOVA), *n* = 6 mice per group. (**D**) The combination of otherwise ineffective doses of Carba1 and PTX inhibits the growth of HeLa cells xenografted in mice. When the tumors have reached a volume of about 200 mm^3^, mice were treated with PTX (3 mg/kg), Carba1 (60 mg/kg), the vehicle, or the combination of PTX (3 mg/kg) and Carba1 (60 mg/kg). Tumor growth was monitored with a sliding caliper. Error bars = SEM. * *p* < 0.001 compared vehicle (Student’s *t*-test), *n* = 8 mice per group. Arrows indicate the onset of the treatments.

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
