# Peer review of "Two Antagonistic Microtubule Targeting Drugs Act Synergistically to Kill Cancer Cells"

_cancers, 2020, doi:10.3390/cancers12082196_

Round 1

Reviewer 1 Report

Herein, Peronne et al. discuss the activity of two microtubule targeting drugs that synergize when exposed to cancer cells. This is an interesting study with some issues that should be addressed.

  1. The majority of the study is performed on a single HeLa cell line, making it difficult to determine what responses may be expected to be general, and which may be cell-line specific. Not sure why the authors did not include 4T1 cells, used in animal studies, for comparison.
  2. The results with the use of non-cancerous cell line are missing to prove the specificity of the drug combination and safety of the proposed treatments
  3. The authors claim the synergy in vivo results but this proves is not anyhow quantified. Please add the quantification.
  4. There is no prove that the treatment is really save for the animals are this claim is based only on the observed weight loss, or rather its lack.
  5. How many times the in vivo experiment was performed?
  6. In vitro assays are preformed mostly only twice therefore the statistical analysis is not adequately used.
  7. The authors claim: “Our results pave the way for new therapeutic strategies, based on the combination of low 90 doses of MT targeting agents with opposite mechanisms of action”. There is a recent paper on the optimized drug combination (https://pubmed.ncbi.nlm.nih.gov/31652588/?from_term=nowak-sliwinska&from_sort=date&from_pos=6), not mentioned in this manuscript, where four drugs were combined and displayed synergistic activity via prevention of spindle pole clustering. It is known that the treatment with a combination of two drugs would much faster lead to the resistance induction than in the case of multidrug combination. The authors should perform an experiment of long-term treatment of their cells to determine when/ if such resistance would occur as compared to TX only.

Author Response

Reviewer #1

Herein, Peronne et al. discuss the activity of two microtubule targeting drugs that synergize when exposed to cancer cells. This is an interesting study with some issues that should be addressed.

We thank the reviewer for his constructive comments and we have revised our manuscript accordingly. We have addressed all of the reviewer's comments.

1. The majority of the study is performed on a single HeLa cell line, making it difficult to determine what responses may be expected to be general, and which may be cell-line specific. Not sure why the authors did not include 4T1 cells, used in animal studies, for comparison.

We have indeed shown the synergy in murine 4T1 cells, a breast cancer cell line (supplementary figure S2A).

Moreover, with the goal to identify the mechanism of action of the synergy we also have initially screened a yeast cDNA library for rescue of the synthetic lethal combination of the two drugs. Unlike human cells, yeast cells have many efflux pumps which make them partially resistant to drugs. In addition, the sequence of -tubulin is different and does not allow PTX binding. We therefore used a yeast strain mutated on 7 efflux pumps and transformed with a DNA fragment containing the -tubulin gene (TUB2) mutated on 5 amino acids thus allowing PTX binding. Using this modified yeast strain we observed that Carba1 (25μM) and PTX (25μM) had no effect on the growth of yeast cells when applied separately, but showed complete growth inhibition when applied together (see figure below). The synergistic effect of the combination Carba1/PTX, observed in human and murine cells, is therefore also valid in yeast, which suggests that the mechanism of action of Carba1 is conserved from yeast to humans.

Interestingly, tubulin also came up as a possible rescue candidate, reinforcing the proposed mechanism of action. We have so far not included these results in the current manuscript, but could do so by adding them as a supplementary figure, if the reviewer finds them useful.

2. The results with the use of non-cancerous cell line are missing to prove the specificity of the drug combination and safety of the proposed treatments

The proposed mechanistic explanation of the observed synergy mechanism is that catastrophe induction by Carba1 promotes paclitaxel binding to microtubule ends. As shown above, such a mechanism can be generalized to many cell types and it is most probable that the combination would also affect non-cancerous cell lines. As stated in the discussion (P15, L448), given this mode of action of Carba1, we can bet that the combination should diminish only microtubule independent PTX adverse events.

3. The authors claim the synergy in vivo results but this proves is not anyhow quantified. Please add the quantification.

By definition, a synergistic effect is greater than the sum of the individual effects of each drug, which is what we observe in vivo. Indeed, at doses where neither paclitaxel nor Carba1 have any effect, their combination has an important impact on tumor growth. This was quantified by the measure of the tumor volumes, which is shown in Figure 7. In order to be clearer, we now specify the values of tumor volumes in the text (P13, L397-399).

4. There is no prove that the treatment is really save for the animals are this claim is based only on the observed weight loss, or rather its lack.

Our overall aim was to establish the proof of concept that the combination could be more effective than each drug administrated separately. This seems important to us before spending major efforts to optimize the Carba1 compound.

So far we are not within the frame of the development of a new drug, which requires in-depth toxicity analyzes. This point is clearly stated in the discussion. We plan to conduct an in-depth toxicity study, including neurotoxicity once the compound is optimized.

We have however replaced the verb "indicating"  (P12 L371) by "suggesting", to tune down our conclusion and avoid possible misinterpretation.

5. How many times the in vivo experiment was performed?

We have performed once two different in vivo experiments (orthotopic grafts of 4T1 cells and xenografts of HeLa cells). In each of these experiments 6 to 8 mice per treatment were used. Such an experimental design is usually considered to be sufficient to establish a proof of concept and is in accordance with the principles of the 3Rs (Replacement, Reduction and Refinement).

6. In vitro assays are preformed mostly only twice therefore the statistical analysis is not adequately used.

Maybe the reviewer missed the fact that all the experiments have been repeated 3 times except experiments of Figure 6, as recapitulated below.

  • Figures 1A & 1B: not applicable (NA); Figures 1C& 1D: 3 separate experiments;
  • Figure 2A-C: 3 separate experiments;
  • Figure 3A:NA, 3B: NA, it is an illustrative analysis that gives convergent results with figure 4;
  • Figure 4A-C: 3 separate experiments;
  • Figure 5A: NA, illustrates and strengthens previous results, Figure 5B-C: 3 separate experiments; Figure 5D-E: NA
  • Figure 6: These in vitro experiments were indeed performed only twice (N = 2). However, in each case, 3 individual movies were collected. All data points (n) from 6 individual movies were combined and used for statistical analysis. We have modified the figure legend to clarify that point.
  • Figure 7: in vivo experiments, which were conducted only once, as stated above, but with sufficient animals per group to perform the appropriate statistical analysis;
  • Figure S1: 3 separate experiments
  • Figure S2A: 3 separate experiments, S2B-C: NA
  • Figure S3: NA

7. The authors claim: “Our results pave the way for new therapeutic strategies, based on the combination of low  doses of MT targeting agents with opposite mechanisms of action”. There is a recent paper on the optimized drug combination (https://pubmed.ncbi.nlm.nih.gov/31652588/?from_term=nowak-sliwinska&from_sort=date&from_pos=6), not mentioned in this manuscript, where four drugs were combined and displayed synergistic activity via prevention of spindle pole clustering. It is known that the treatment with a combination of two drugs would much faster lead to the resistance induction than in the case of multidrug combination. The authors should perform an experiment of long-term treatment of their cells to determine when/ if such resistance would occur as compared to TX only.

The analysis of the occurrence of resistance upon the combined administration of Carba1 and PTX is not in the focus of the present manuscript. We agree that it is a very interesting point that we plan to analyze in further studies, using a pharmacologically optimized compound. We plan 1) to analyze the effect of the optimized combination on PTX resistant cell-lines and on their non-resistant counterparts and 2) to perform the reviewer's suggested long-term experiments on HeLa cells.

Reviewer 2 Report

This is a very interesting study that convincingly shows that Carba1, a compound selected by the authors from a chemical library of 8,000 molecules on the basis of its ability to sensitize cells to the anticancer drug paclitaxel, is a microtubule destabilizing agent that targets the colchicine binding-site of tubulin, hence favouring the accumulation of paclitaxel (a microtubule stabilizer) inside microtubules, thereby explaining the observed synergy between both compounds and the apparent discrepancy by which a microtubule destabilizing agent can synergize a microtubule stabilizer. In addition, the author also observed in vivo this synergistic effect of paclitaxel and Carba1 in xenografted mice, thus strongly suggesting that the combination of both these compounds represents a new therapeutic strategy for cancer that may have less side effects than paclitaxel alone.

In my view, this manuscript could be accepted in its present form.

Author Response

We thank the reviewer for taking the time to read and analyze our manuscript.

Reviewer 3 Report

The manuscript by Peronne et al. presents a very interesting translational study dealing with an important unmet medical need, namely the design of new therapies to increase tumor sensitivity to paclitaxel. Authors report here the discovery of a novel microtubule-destabilizing drug that synergizes with paclitaxel to kill cancer cells and may be useful in combination for anti-cancer therapy.

The experiments are carefully performed and adequately discussed. The study includes a wide range of elegant methods to identify and characterize the new compound: drug screening, in vitro binding studies, time-lapse, microtubule dynamics, cell biology and in vivo validation in two different mice models. The work is of interest and results are of good quality, with potential important therapeutic perspectives.

A few points however should be addressed :

How did the authors select 8000 compounds out of 19.000 of the library for drug screening? Did they identify other carba1-chemically related compounds with similar PTX-potentiating effects? It would be useful to provide a supplemental table showing all 76 identified hits.

What is the fate of cells treated with carba1 at 12 µM, a dose that delays metaphase and yet does not promote apoptosis? What is the consequence of carba1 and carba/PTX treatment on multipolar spindles and other abnormal mitotic figures that are induced by PTX?

In vivo 4T1 tumors grow more slowly than tumors formed by HeLa cells injection. What is the difference in proliferation rate and IC50 of the drugs for the two cell lines in vitro? Could the difference in tumor growth rate account for efficient drug combination in Hela compared with 4T1 model? This is an important point that authors should consider and discuss regarding the type of human cancers that would benefit from combined therapy. As an example, high grade ER-negative breast tumors are known to be more responsive to cytotoxic chemotherapy.

Different experimental protocols were used for in vivo studies of 4T1 and HeLa cells, but this is never clearly stated in the results nor discussion and this brings some confusion to the conclusions. In Fig.7, the day of drug administration should be indicated. What could be the consequence of these different experimental procedures on drug combination efficiency?

Minor points

- It is kind of difficult to distinguish between the different curves in Fig 1C, Fig.7 and supplemental Figures. Please use bigger dot symbols or use different colors, as in Fig 5B.

- What is the method used to evaluate Carba1 effects on a panel of protein kinases in vitro (Table S5)?

- What is the effect of carba1 alone at different doses on cell viability in 4T1 cells (missing control in Suppl Fig S2A)

- How do the authors explain that a high PTX dose of 8 mg/kg is highly toxic and killed the mice in the 4T1 exp but not in the HeLa exp?

- The dot symbol for vehicle treatment is missing in Fig.7B

- page 9, line 243 : MT growth (instead of MT growth length)

Author Response

Reviewer #3

The manuscript by Peronne et al. presents a very interesting translational study dealing with an important unmet medical need, namely the design of new therapies to increase tumor sensitivity to paclitaxel. Authors report here the discovery of a novel microtubule-destabilizing drug that synergizes with paclitaxel to kill cancer cells and may be useful in combination for anti-cancer therapy.

The experiments are carefully performed and adequately discussed. The study includes a wide range of elegant methods to identify and characterize the new compound: drug screening, in vitro binding studies, time-lapse, microtubule dynamics, cell biology and in vivo validation in two different mice models. The work is of interest and results are of good quality, with potential important therapeutic perspectives.

We thank the reviewer for his constructive comments and we have revised our manuscript accordingly. We have addressed all of the reviewer's comments.

A few points however should be addressed:

How did the authors select 8000 compounds out of 19.000 of the library for drug screening?

The 8,000 compounds have been extracted from the chemical library using a clustering method in order to adapt the size of the panel to the capacity of the screening, while respecting the structural diversity of the collection. We have now clarified that point in the Material and Methods section (P16L508-509).

Did they identify other carba1-chemically related compounds with similar PTX-potentiating effects? It would be useful to provide a supplemental table showing all 76 identified hits.

The 76 identified hits belong to several chemically different families and it would be uninformative to show all these compounds without having been validated in a secondary screen.

We indeed identified other carba1- chemically related derivatives, which strengthens the interest of this family. Following the reviewer's suggestion, we now add a table (Table S3) that shows these additional compounds. We tested the effect of some of these compounds on HeLa cell viability and show these additional results now also in Table S3.

What is the fate of cells treated with carba1 at 12 µM, a dose that delays metaphase and yet does not promote apoptosis?

As shown in Figure 1C (value of viability at the ordinate, corresponding to 0 nM of PTX and 12µM of Carba1), and Figure 2A, although Carba 1 at 12 µM delays metaphase, cells can proliferate and such a concentration has only a minimal effect on cell viability.

What is the consequence of carba1 and carba/PTX treatment on multipolar spindles and other abnormal mitotic figures that are induced by PTX?

Table S5 shows the quantification of the fate of cells treated with different concentrations of Carba1 and/or PTX.  A high concentration of Carba1 (25 µM) induced a prometaphase arrest that was followed by apoptotic death, which was visible on the 20-hour movie (Movie S2).

The Carba1/PTX combination induced a mitotic delay and multipolar spindles similar to what is observed with PTX 5 nM (Movie S3). On the duration (12 hours) of the movie, we were not able to observe apoptosis. However, flow cytometry analysis (Figure 4), conducted over longer periods of time showed that the cells become multinucleated and die. Because of the mechanism of action of Carba1, we assume that cell death mechanisms involved in the Carba1/PTX response are similar to those involved in the PTX response  (please, see for review Shi and Mitchison, PMID: 28249963), but we did not fully demonstrate that point.

In vivo 4T1 tumors grow more slowly than tumors formed by HeLa cells injection. What is the difference in proliferation rate and IC50 of the drugs for the two cell lines in vitro? Could the difference in tumor growth rate account for efficient drug combination in Hela compared with 4T1 model? This is an important point that authors should consider and discuss regarding the type of human cancers that would benefit from combined therapy. As an example, high grade ER-negative breast tumors are known to be more responsive to cytotoxic chemotherapy.

The proliferation rate of 4T1 cells in vitro is similar to the HeLa cell proliferation rate. As explained in the text and as shown on Figure 1C and Figure S2A, these cells differ in their sensitivity to PTX: the IC50 of PTX is 1.5 nM for HeLa cells and 90 nM for 4T1 cells. It is rather complicated to compare the tumor growth rate in vivo in these two models, as one is an allogeneic model (4T1), with cells (20,000) injected into the mammary fat pad, while the other is a xenograft-based model (HeLa), with cells (10 7) injected into the flank.

We agree with the reviewer that it would be important to know which kind of human cancer would benefit from a combined therapy. We plan to conduct this type of analysis as part of a program to develop the Carba1 compound (which is now patented) and its derivatives.

Following the reviewer's suggestion, we have added a paragraph in the discussion (P15, L459-470), to discuss the different tumor responses to the combination observed in the two in vivo models.

Different experimental protocols were used for in vivo studies of 4T1 and HeLa cells, but this is never clearly stated in the results nor discussion and this brings some confusion to the conclusions. In Fig.7, the day of drug administration should be indicated. What could be the consequence of these different experimental procedures on drug combination efficiency?

We agree with the reviewer that the different experimental protocols of in vivo studies are not clearly explained, especially in the figures. We have modified the figure 7 accordingly, indicating the onset of the treatments.  Moreover, in the paragraph added to the discussion, we now also discuss the different experimental protocols. It is true that the different the different route of injection, for instance, could have an influence on drug availability at the level of the tumor, or on drugs half-lives.

Minor points

  • It is kind of difficult to distinguish between the different curves in Fig 1C, Fig.7 and supplemental Figures. Please use bigger dot symbols or use different colors, as in Fig 5B.

We now provide new figures, with different colors.

  • What is the method used to evaluate Carba1 effects on a panel of protein kinases in vitro (Table S5)?

We thank the reviewer for this observation. Indeed, we have forgotten to provide this information. This is now done, in the materials and methods section, P16, L544-548.

  • What is the effect of carba1 alone at different doses on cell viability in 4T1 cells (missing control in Suppl Fig S2A)

This control is not missing. It corresponds to the value of viability at the ordinate, corresponding to 0 nM of PTX and the different doses of Carba1. As this information is not immediately evident we have added an explanation in the legend, as well as in the legend of Figure 1C.

  • How do the authors explain that a high PTX dose of 8 mg/kg is highly toxic and killed the mice in the 4T1 exp but not in the HeLa exp?

As stated in the material and methods section (and now also in the discussion) the experimental protocols are different: in the 4T1 experiment, where the dose of 8 mg/kg of PTX was found to be highly toxic, PTX was administrated (i.p.) every day; in the HeLa cell experiment, it was administrated (i.v.) every two days. We believe that these different routes of injection, as well as the different schedules explain the difference in the PTX toxicity.

  • The dot symbol for vehicle treatment is missing in Fig.7B

This figure is now in color.

- page 9, line 243 : MT growth (instead of MT growth length)

We apologize for this mistake, which is now corrected.

Round 2

Reviewer 1 Report

The authors partially answered my concerns, being a bit in hurry, as I would guess because of re-submission deadline policy. This can be easily postponed, especially if this reviewer asked for some additional experiments.

I am still not fully satisfied with the answers. It seems that they disagree with my comments and do not want to address them. Nevertheless, this would strongly increase the importance of the study.

I can not accept the explanations like “ we can bet that…”. In the drug development world, any synergistic drug interaction may lead to simultaneous synergy in toxicity. Therefore, the proof that is not the case in non-cancerous cells is of major importance, especially that paclitaxel, is rather toxic.

Point on synergy: the authors does not need to explain the reviewer what synergy is by definition. They should quantify it. There are multiple tools to do that.

This might be country dependent, but in my experience performing one the experiments with mice is not sufficient to perform the correct statistical design. With the whole respect of the 3R rule, that we all apply. It would be good if at least the authors include the power analysis for in vivo studies.

Another problem is in in vitro experiments – e.g. Figure 6. If the experiments were done twice (two independent biological replicates) the authors should not take in the analysis the 6 points (movies), but only two.

Author Response

The authors partially answered my concerns, being a bit in hurry, as I would guess because of re-submission deadline policy. This can be easily postponed, especially if this reviewer asked for some additional experiments.

It is true that we found it difficult to carry out the additional experiments requested by the reviewer within the 10-day period granted by the editor. As suggested by the reviewer, we requested an extension, which was granted.

I am still not fully satisfied with the answers. It seems that they disagree with my comments and do not want to address them. Nevertheless, this would strongly increase the importance of the study.I can not accept the explanations like “ we can bet that…”. In the drug development world, any synergistic drug interaction may lead to simultaneous synergy in toxicity. Therefore, the proof that is not the case in non-cancerous cells is of major importance, especially that paclitaxel, is rather toxic.

As suggested by the reviewer, we have analyzed the toxicity of the compound Carba1 alone and in combination with Paclitaxel on a normal cell line (immortalized RPE-1 human cells). We found that Carba1 alone was not toxic on this cell line and exerts a synergistic action only for PTX concentrations that affect cell viability. These additional data are presented on Figures S3 and S4 and detailed in the main text Page 5 (L 164-170).

Point on synergy: the authors does not need to explain the reviewer what synergy is by definition. They should quantify it. There are multiple tools to do that.

We have used the combination index method of Chou and Talalay to quantify the synergy of Carba 1 and PTX on Hela cells (Figure S2), and 4T1 cells (Figure S6). These analyses - which are presented in the main text page 3 (L 111-119) and page 12 (L339)- confirmed the synergistic action of the PTX - Carba1 combination.

We have added a paragraph in the materials and methods section (page 16, L 541-553), explaining how we have proceeded to conduct these analyses, and two additional references (Chou and Talalay, 1984; Chou, 2006).

This might be country dependent, but in my experience performing one the experiments with mice is not sufficient to perform the correct statistical design. With the whole respect of the 3R rule, that we all apply. It would be good if at least the authors include the power analysis for in vivo studies.

We have checked that the statistical tests used for comparison of the groups were appropriate for the number of repeats (6-8 mice per group, depending of the experiment). In the experiment that shows that a synergy can be observed between Carba1 and PTX (Figure 7D, 8 mice per group), we have used a t-test to compare the volumes of the tumors at the end of the experiment, after checking the normality of the data and the homogeneity of the variances. We have corrected the figure's legend accordingly and added a paragraph (which includes the power analysis of the test) in the materials and methods section (Page 19, L 678-686).

Another problem is in in vitro experiments – e.g. Figure 6. If the experiments were done twice (two independent biological replicates) the authors should not take in the analysis the 6 points (movies), but only two.

We apologize for misunderstanding: The statistics was not based on n=6 movies but on the number of microtubules analyzed in each condition, as is habitual in the field. We did not use n=6 for any of the analyses. We do report that all experiments were performed twice and the data are in good agreement between the two biological replicates for each condition. We have modified the legend in Figure 6 to clarify this point.

We thank the reviewer for his (her) constructive remarks.

The additional experiences and analyses we have carried out on his advice have strengthened the conclusions and we are particularly grateful to him (her) for this.

Round 3

Reviewer 1 Report

The authors responded to my comments. the paper can be accepted in the current format.